# Non-Targeted Metabolomics Approach Revealed Significant Changes in Metabolic Pathways in Patients with Chronic Traumatic Encephalopathy

**DOI:** 10.3390/biomedicines10071718

**Published:** 2022-07-15

**Authors:** Jinkyung Lee, Suhyun Kim, Yoon Hwan Kim, Uiyeol Park, Junghee Lee, Ann C. McKee, Kyoung Heon Kim, Hoon Ryu, Jeongae Lee

**Affiliations:** 1Center for Advanced Biomolecular Recognition, Korea Institute of Science and Technology (KIST), Seoul 02792, Korea; jin985656@gmail.com (J.L.); yoon.yh89@gmail.com (Y.H.K.); 2Department of Biotechnology, Graduate School, Korea University, Seoul 02841, Korea; khekim@korea.ac.kr; 3Brain Science Institute, Korea Institute of Science and Technology (KIST), Seoul 02792, Korea; shkimail@kist.re.kr (S.K.); uypark@kist.re.kr (U.P.); 4Boston University Alzheimer’s Disease Research Center (BUADRC), Department of Neurology, Boston University School of Medicine, Boston, MA 02118, USA; junghee@bu.edu (J.L.); amckee@bu.edu (A.C.M.)

**Keywords:** chronic traumatic encephalopathy (CTE), non-targeted metabolomics, astrocyte activation, catecholamines, tyrosine metabolism, phenylalanine metabolism

## Abstract

Chronic traumatic encephalopathy (CTE) is a neurodegenerative disease that is frequently found in athletes and those who have experienced repetitive head traumas. CTE is associated with a variety of neuropathologies, which cause cognitive and behavioral impairments in CTE patients. However, currently, CTE can only be diagnosed after death via brain autopsy, and it is challenging to distinguish it from other neurodegenerative diseases with similar clinical features. To better understand this multifaceted disease and identify metabolic differences in the postmortem brain tissues of CTE patients and control subjects, we performed ultra-high performance liquid chromatography–mass spectrometry (UPLC-MS)-based non-targeted metabolomics. Through multivariate and pathway analysis, we found that the brains of CTE patients had significant changes in the metabolites involved in astrocyte activation, phenylalanine, and tyrosine metabolism. The unique metabolic characteristics of CTE identified in this study were associated with cognitive dysfunction, amyloid-beta deposition, and neuroinflammation. Altogether, this study provided new insights into the pathogenesis of CTE and suggested appealing targets for both diagnosis and treatment for the disease.

## 1. Introduction

Chronic traumatic encephalopathy (CTE) is a neurodegenerative disease that is observed in people with a history of repetitive head trauma. This disease was first recognized as “punch drunk” and “dementia pugilistica” to describe the common neurological and neurobehavioral issues found in professional boxers due to a series of traumatic head blows during their boxing careers [1]. Similar clinical signs and symptoms were reported in athletes of other contact sports including American football [2], wrestling [3], rugby [4], soccer [5], ice hockey [5], as well as in non-sports-related personnel including war veterans [6], circus performers [7], and victims of domestic abuse [8].

CTE has a wide range of clinical symptoms that negatively affect cognition, behaviors, and mood. In mild cases, common symptoms are attention and concentration difficulties, depression, headache, mood swings, and short-term memory loss. In serious cases, cognitive impairment, dementia, disturbance, executive dysfunction, explosive behavior, gait disturbance, language impairment, parkinsonism, suicidality, and visuospatial difficulties have been reported [1,5,9]. Despite these symptoms, CTE cannot be diagnosed in a living person today, but only by evaluating the brain tissues after death.

According to the most recent National Institute of Neurological Disorders and Stroke/National Institute of Biomedical Imaging and Bioengineering (NINDS/NIBIB) consensus meeting [10], the minimum requirement for CTE diagnosis is the presence of a single pathognomonic lesion in the cortex: “Hyperphosphorylated tau (p-tau) aggregates in neurons, with or without thorn-shaped astrocytes, at the depth of a cortical sulcus around a small blood vessel, deep in the parenchyma, and not restricted to the subpial and superficial region of the sulcus.” To determine the severity of the disease, the presence of neurofibrillary tangles (NFTs) in 10 different regions of the brain including the side and crest of the gyrus, superficial cortical laminae, CA4 and CA2 of hippocampus, entorhinal cortex, amygdala, thalamus, mammillary body, and cerebellar dentate nucleus are evaluated. The disease is considered mild (“Low CTE”) or severe (“High CTE”) depending on the number of NFTs in the brain [10]. “Low CTE” is equivalent to stages I and II and “High CTE” is equivalent to stages III and IV of the pathological stages proposed by McKee et al. [5,10].

Though many studies recognized CTE as a distinct disease [5,9,11], distinguishing it from relatively well-understood neurodegenerative diseases such as Alzheimer’s disease (AD), Parkinson’s disease (PD), and amyotrophic lateral sclerosis (ALS), it is still a major diagnostic challenge. Many efforts have been devoted to diagnosing CTE in a living person by observing structural and physiological changes via neuroimaging techniques, namely magnetic resonance imaging (MRI), positron emission tomography (PET), and flortaucipir (FTP) [12,13,14,15,16]. However, not only does the significance of the findings of CTE remain to be elucidated since most studies focus on traumatic brain injury (TBI) rather than CTE [13,14], the imaging technologies also lack specificity and sensitivity for diagnosing CTE in a living person [12].

Since CTE has a complex, multifaceted nature similar to other neurodegenerative diseases, metabolomics can be a useful tool for studying the dynamic interactions between different types of molecules in multiple biological pathways that are specific to CTE pathophysiology [17]. In this study, we employed non-targeted metabolomics to detect a wide range of metabolites in the postmortem brain tissues of CTE patients and control subjects. With the non-targeted metabolomics approach, we discovered a variety of compounds involved in brain functions and uncovered metabolic characteristics of CTE, which can advance our knowledge and help with early diagnosis of the disease.

## 2. Materials and Methods

### 2.1. Chemicals and Reagents

HPLC-grade acetonitrile (ACN) and ethanol were purchased from Burdick & Jackson (Burdick & Jackson, Muskegon, MI, USA). The following buffer solutions were purchased from Sigma-Aldrich (Sigma-Aldrich, St. Louis, MO, USA): formic acid, ammonium acetate, and phosphate-buffered saline (PBS). A milli-Q purification system (Millipore, Bedford, MA, USA) was used to obtain deionized water for making aqueous solutions.

### 2.2. Human Brain Samples

Neuropathological processing of control and CTE human brain samples were performed according to the procedures previously established for the Chronic Traumatic Encephalopathy (CTE) Center. Stages of CTE were determined according to Dr. McKee’s criteria [18], which is based on the density and regional deposition of p-tau pathology, and the criteria for pathological diagnosis was adopted from and refined by NINDS/NIBDB. Next of kin provided informed consent for participation and brain donation. The study was performed in accordance with the institutional regulatory guidelines and principles of human subject protection in the Declaration of Helsinki. Clinical and demographic information of postmortem brain tissues from control subjects and CTE patients is summarized in Appendix A. McKee et al. (2015) had reported that the temporal lobe exhibits an increase in phosphorylated Tau (p-Tau) in stage IV of CTE while the region does not show prominent signals of p-Tau at the early stage of CTE [18]. Accordingly, it is proposed that the temporal cortex reflects the pathological progression of CTE through the spreading of p-Tau after the head injury. In this context, we chose to study the temporal cortex to determine whether it shows metabolomic changes, similar to the progressive pathological change in response to repetitive brain trauma.

### 2.3. Sample Preparations

In total, 100 mg of frozen postmortem brain tissue was extracted using 500 μL of degassed extraction solvent (ethanol/10 mM PBS; 85:15, *v*/*v*) in a Safe-Lock tube (Eppendorf, Hamburg, Germany) containing ceramic beads. The purpose of using degassed extraction solvent was to eliminate oxygen in the solvents and prevent oxidation of thiols or antioxidant metabolites that may be present in the samples [19]. After the extraction, samples were homogenized at 25 Hz for 5 min using a TissueLyser (Qiagen, Hilden, Germany). The samples were then centrifuged at 12,000× *g* for 10 min at 4 °C. Then, 100 μL of supernatant was collected and transferred into another clean test tube and evaporated to dryness under a gentle stream of nitrogen. A total of 100 μL of 0.1% formic acid in 5% ACN/95% water solutions was added to reconstitute the residue and 5 μL of each sample was injected to UPLC-LTQ-Orbitrap MS for analysis.

### 2.4. LC-MS Analysis

LC-MS analysis was performed using an Ultimate 3000 UHPLC system coupled with an LTQ-OribitrapVelos Pro hybrid mass spectrometer (Thermo Fisher Scientific, Waltham, MA, USA). Prepared samples were kept at 4 °C in an autosampler and injected onto an Acquity^TM^ UPLC BEH C18 (2.1 × 100 mm, 1.7 μm) column at 40 °C with a flow rate of 0.35 mL/min for reversed-phase separation in both positive (ESI+) and negative (ESI−) ion modes. The mobile phases were solvent (A) 0.1% formic acid in 5% ACN/95% water and solvent (B) 0.1% formic acid in 95% ACN/5% water. The total run time was 14 min with gradient elution as described in Appendix A. The same elution program was used for both ESI+ and ESI− to detect metabolites with the same retention time in both ion modes. Samples were analyzed using Fourier transform mass spectrometry (FTMS) full scan mode with resolving power 100,000 at *m*/*z* ranges 50–1200 in centroid mode. The capillary temperature was 320 °C. The spray voltages were 5 and 4 kV for ESI+ and ESI−, respectively.

### 2.5. Validation

A pooled quality control (QC) sample was prepared by thoroughly mixing small aliquots of each sample for system stability and repeatability validation. In total, 10 QC samples were injected before the sample analysis in order to equilibrate the system to acquire reproducible data [20]. The solvent blank and QC sample were injected in every 9 samples to validate the repeatability within an analytical batch.

### 2.6. Data Processing and Statistical Analysis

The raw data acquired from LC-MS were processed by Thermo Scientific SIEVE software v2.1 with “Small molecule”, “Chromatographic Alignment and Framing”, and “Non-differential single class analysis” options [21]. As a pretreatment process, all data was scaled using the Pareto scaling method to diminish the mask effect from the abundant metabolites and achieve better results in statistical analyses [22].

For the univariate analysis, fold-change analysis and *t*-test were performed using MetaboAnalyst 5.0 (www.metaboanalyst.ca, accessed on 7 February 2022). Changes in metabolite concentrations were determined by fold-change ratios, and the significant differences between the means of the healthy control group and the CTE group were determined by *p*-value using a *t*-test and Wilcoxon rank-sum test [23]. For multivariate analysis (MVA), SIMCA software v16 (Umetrics, Umeå, Sweden) was used to perform a principal component analysis (PCA) and orthogonal projection to latent structures–discriminant analysis (OPLS-DA). First, PCA was performed to observe the sample pattern and detect outliers without considering the group membership [24]. Next, OPLS-DA was performed to optimize the group separation between the groups and identify the metabolites contributing the most to the separation based on the Variable Importance in Projection (*VIP*) values [25]. Metabolites with *VIP* > 1 and *p*-value < 0.05 were considered significant.

### 2.7. Metabolite Identification and Pathway Analysis

For the putative identification of selected significant metabolites, experimentally determined accurate masses were submitted and matched to the compounds in databases using the web server MassTRIX (www.masstrix.org, accessed on 24 January 2022). The identification was performed with the following parameters: scan mode for each positive and negative ionization; maximum error of 0.05 Da; “KEGG/HMDB/LipidMaps without isotopes” databases; and *Homo sapiens* (human) as the reference organism [26]. Putatively identified metabolites were presented on the KEGG pathway map to observe their position in metabolic pathways using the KEGG Mapper-Search and Color Pathway tool [27]. Then, pathway analysis was conducted using MetaboAnalyst 5.0 to identify the most significantly affected pathways by the selected metabolites. The interaction between the significant metabolites in significantly altered pathways were visualized by performing network analysis using Cytoscape 3.8.2.

### 2.8. RNA Sequencing and Analysis

RNA seq data retrieved from our published data was analyzed [28]. The RNA sequencing data are available under the European Nucleotide Archive (ENA) accessions no. ERP110728. In brief, the normal and CTE brain samples were prepared for sequencing using the Illumina TruSeq RNA sample preparation kit (Illumina, San Diego, CA, USA) according to the manufacturer’s instructions and sequenced on a HiSeq 2000 platform. The 101 bp sequenced paired-end reads were mapped to the hg19 reference human genome using the STAR 2-pass method. We used HTSeq to count the reads aligned to each gene based on the Ensembl gene set. We excluded samples that failed in the library preparation or sequence process. We also excluded samples with fewer than 10 million reads sequenced. Overall, 10 CTE subjects and 9 control subjects were examined. The normalized read counts were applied to principal component analysis or clustering analysis, which was conducted through R and Cluster 3.0 and visualized via Java Treeview (Version 1.1.6r4).

### 2.9. Immunohistochemistry (IHC)

Neuropathological processing of normal and CTE postmortem brain samples was performed using procedures previously established by the Boston University Alzheimer’s Disease Center (BUADC) [29]. Briefly, endogenous peroxidase activity in the deparaffinized sections was inactivated by the solution (BLOXALL, Vector Laboratories, Burlingame, CA, USA, SP-6000) for 10 min. The tissue sections were blocked with blocking solution (5% BSA (BSAS 0.1; Bovogen Biologicals, Keilor East, VIC, Australia) in TBST) for 1 h and then further incubated with specific antibodies for SHMT2 (1:200, ab224427; Abcam, Waltham, MA, USA) for 24 h. Then the sections were incubated in the biotinylated IgG (1:200, DI-1594; Vector Laboratories, Burlingame, CA, USA), and with 3% normal goat serum in TBS for 90 min. After washing, the sections were incubated in prepared VECTASTAIN Elite ABC reagent (PK-6100; Vector Laboratories, Burlingame, CA, USA) for 30 min. Then, the sections were incubated in peroxidase substrate solution (DAB, 3,3′-diaminobenzidine tetrahydrochloride hydrate, D5637; Sigma, St. Louis, MO, USA) for 3 min. Next, the sections were incubated with astrocyte maker GFAP (1:400, ab7260; Abcam, Waltham, MA, USA) in the same blocking solution for 24 h. The sections were then incubated with alkaline phosphatase reagent (ImmPRESS-AP REAGENT, MP-5041; Vector Laboratories, Burlingame, CA, USA) for 2 min and rinsed and mounted. Images were acquired using light microscopy (BX63, Olympus, Tokyo, Japan). Then, they were analyzed using Fiji software (NIH, Bethesda, MD, USA) with color deconvolution which is a function of separation of IHC images into DAB (brown) and alkaline phosphatase (red) signals. To analyze the co-localization of GFAP and SHMT2, images were applied to the intensity profile in Fiji software (NIH, Bethesda, MD, USA) [30].

## 3. Results

### 3.1. Metabolic Profiling of Postmortem Brain of the CTE Patients

Based on the non-targeted metabolomics analysis, a total of 458 variables were detected in the postmortem brain of the 10 CTE patients and 9 normal subjects. For the method validation, a total of 13 QC samples were used to evaluate the repeatability of the metabolic profiling. The clustering and a clear separation of QC samples from the rest of the experimental samples in the unsupervised PCA plots (both positive and negative ion modes) demonstrated that the data was acquired with a good repeatability (Appendix A). As part of the quality assurance (QA) procedure, variables detected in fewer than half of the QC samples with RSD more than 30% were considered unreliable, and so omitted from future multivariate analysis [20,31]. In this study, it was observed that more than half of the variables detected in QC samples in both positive (65.0%) and negative (69.3%) modes showed RSD values below 30% (Appendix A). Taken together, a reliable set of data was obtained in this study, and a total of 251 variables were analyzed further to evaluate their significance in distinguishing the CTE group from the control group.

### 3.2. Multivariate Analysis

The multivariate analysis was performed to observe the group pattern and identify metabolic characteristics of the postmortem brain of the CTE patients. Although not completely separated, clustering of data with the evident separation tendency between the CTE and control groups was obtained in both positive (*R2X* = 0.852 and *Q2* = 0.549) and negative (*R2X* = 0.517 and *Q2* = 0.061) ion modes, as shown in the score plot of our PCA models (Figure 1A,B). Following that, supervised OPLS-DA was used to optimize group separation and extract metabolites capable of distinguishing between the CTE and the control groups. In our OPLS-DA model, a complete group separation was observed in both positive (*R2X_(cum)_* = 0.450 and *Q2_(cum)_* = 0.337) and negative (*R2X_(cum)_* = 0.543 and *Q2_(cum_*_)_ = 0.708) ion modes (Figure 1C,D). In targeted metabolomics, R2 and Q2 values above 0.4 were considered as a robust model with a strong predictive power [32]. Given that our model was based on non-targeted metabolomics, our OPLS-DA model in both ion modes demonstrated a high explained variance with a moderate predictive ability, and 69 metabolites showed *VIP* > 1.

### 3.3. Identification of Significant Metabolites

Out of 69 differential metabolites, 27 metabolites showed significant differences between the CTE and healthy control groups with *p*-value < 0.05. The putative identification of significant metabolites (*VIP* > 1 and *p*-value < 0.05) was performed using the accurate mass and all the metabolites except 5 metabolites were identified with the maximum error of 0.05 Da (Table 1). Compared to the control group, the concentrations of 21 metabolites were significantly higher in the CTE group while only 6 metabolites including cyclic GMP, 1-pyrrolidine-5-carboxylic acid, 3,4-dihydroxyphenylacetaldehyde, carbon dioxide, and two unknown metabolites were detected with significantly lower levels in the CTE group (Figure 2).

### 3.4. Metabolic Pathway Analysis

Metabolic pathway analysis was performed in order to investigate the impact of significant metabolites in metabolic pathways. The result showed that a total of eight metabolic pathways were significantly altered (Raw *p* < 0.05, Impact > 0.1) in the CTE patients (Figure 3). The affected pathways include tyrosine metabolism, arginine and proline metabolism, glycine, serine and threonine metabolism, aminoacyl-tRNA biosynthesis, phenylalanine, tyrosine and tryptophan biosynthesis, phenylalanine metabolism, nicotinate and nicotinamide metabolism, and retinol metabolism (Appendix A). The network analysis was performed with the results of the metabolic pathway analysis in order to see the connection between the metabolites. As shown in Figure 4, phenylalanine and tyrosine were involved in the pathways with the highest impact values, namely phenylalanine, tyrosine and tryptophan biosynthesis, phenylalanine metabolism, and tyrosine metabolism. Based on this finding, it is reasonable to conclude that disruptions in these pathways were responsible for the accumulation of phenylalanine, tyrosine, vanylglycol, and norepinephrine in the brains of CTE patients, though the precise mechanism underlying the changes in metabolite concentration requires further investigation.

### 3.5. Alterations of Serine Pathway-Related Gene Expression

Because only a limited number of metabolites were detected in each metabolic pathway, the expression patterns of specific candidate genes involved in significantly altered metabolic pathways were examined to see if expression levels correlated with changes in metabolite concentrations (Figure 5). Importantly, serine pathway-associated genes such as *SHMT1*, *SHMT2,* and *THA1P* were upregulated in CTE patients (Figure 5A). Additionally, catecholamine pathway-associated genes such as *COMT* and *MAOA* were upregulated in CTE patients (Figure 5B). In contrast, tyrosine pathway-associated genes such as *DBD*, *TH,* and *DDC* were downregulated in CTE patients (Figure 5C). A phenylalanine pathway-associated gene such as PAH was slightly increased but it was not significant (Figure 5D). The scheme in Figure 5E was drawn to show that two enzymes such as *SHMT1* in the cytosol and *SHMT2* in the mitochondria are involved in producing serine. *SRR* (serine racemase) expression was slightly increased in CTE patients but it was not statistically significant (*p* = 0.316) (Appendix A). Considering that we used brain tissues of control subjects and CTE patients, it seems likely that cell-type-specific change in SRR expression may be diluted and biased with many cell types in the brain.

### 3.6. Immunoreactivity of SHMT2 in Astrocytes of CTE Patients

Since the serine pathway is linked to the pathological progress of neuronal damage, immunoreactivity of SHMT2 was analyzed to examine whether SHMT2 protein expression is increased in CTE and which cell type is involved in SHMT2 expression in the cortex of CTE patients (Figure 6). Concurrent with the increase in *SHMT2* RNA level, a significant increase in SHMT2 immunoreactivity was detected in glial fibrillary acidic protein (GFAP)-positive reactive astrocytes in the white matter and neurons in the gray matter (Figure 6A,B). Further co-localization analysis showed that the SHMT2 signal was found in the cell soma and processes and strongly co-localized with GFAP-positive reactive astrocytes (Figure 6C).

## 4. Discussion

CTE is undoubtedly a distinct neurodegenerative disease, but many aspects of it, especially the pathophysiology of the disease, are not yet fully understood and clearly defined [33]. Consequently, despite many efforts [12,13,14,15,16], there is still a lack of validated biomarkers to detect and diagnose CTE during life, impeding subsequent discovery of treatment and prevention of CTE [34]. We, therefore, employed LC-MS to identify metabolic profiles from the postmortem brain of CTE patients and control subjects in hopes to provide new insights into the pathophysiology of CTE. To our knowledge, analyzing the postmortem brain of CTE patients using LC-MS is a novel approach to exploring metabolic features of CTE. In this study, we discovered that the concentrations of 27 metabolites in the superior temporal cortex of CTE patients and controls were significantly different, and that the differences in metabolite concentrations resulted in significant abnormalities in eight metabolic pathways.

### 4.1. Astrocyte Activation

One of the neuropathological characteristics of CTE is astrocyte activation, also known as reactive astrogliosis [35]. Astrocyte activation refers to the molecular, cellular, and functional changes in astrocytes in response to all sorts of CNS injuries [36]. Astrocyte activation following TBI can have both neuroprotective and neurotoxic effects on the brain [37], but the findings of our study showed astrocyte-associated neurodegenerative metabolic changes in the brain of CTE patients. 

In the brain tissue of CTE patients, higher levels of serine were detected compared to the control subjects. Specifically, d-serine was elevated, which was determined by measuring the expression level of serine racemase (Appendix A), the biosynthetic enzyme converting l-serine to d-serine [38]. d-serine is a gliotransmitter that regulates synaptic transmission and plasticity by acting as a co-agonist for synaptic *N*-methyl-d-aspartate receptors (NMDARs) [39]. However, one study revealed that TBI disrupts the normal function of astrocytes by increasing the production and release of d-serine in astrocytes, which contributes to the slow excitotoxic synaptic damage and cognitive impairment associated with TBI [40]. Moreover, d-serine at pathological level mediates neuronal cell death in neurodegenerative diseases such as AD and ALS by over-activating NMDAR [39,41]. Previous studies have shown that reactive astrocytes exhibit an expression of SRR in vitro and in vivo [38,42]. Our data also suggests that SHMT is induced in astrocytes of CTE. Accordingly, we propose that the increase in SHMT and SRR can contribute to the elevation of d-serine in reactive astrocytes of CTE. These findings suggest that inhibiting the production and release of d-serine has the potential to be a therapeutic target attenuating synaptic and cognitive damage after TBI and prevent the development of the CTE.

Another metabolic change associated with astrocyte activation in CTE is the increased level of gamma-aminobutyric acid (GABA). GABA is a major inhibitory neurotransmitter and its secretion and metabolism are regulated by astrocytes [43]. Similar to D-serine, abnormal production and release of GABA by reactive astrocytes were reported in neurodegenerative disease and were responsible for memory impairment in AD mice [44]. Since reactive astrocytes were also found in CTE [35], it is possible that elevated GABA levels in CTE are the result of the astrocyte activation. Considering the consequence of the high level of GABA in the brain, a study showed that GABA can exacerbate the brain injury, especially in traumatized neurons, by raising intracellular level of Ca^2+^ [45]. Moreover, it is well understood that increased influx of Ca^2+^ has devastating effects on brain health by impairing neuronal plasticity, activating catabolism of proteins and nucleic acids, synthesizing nitric oxide, and stimulating hyperphosphorylation of tau [46,47,48]. Therefore, GABA may play a role in developing CTE in people with multiple head injuries by increasing neuronal damage by raising Ca^2+^ in the brain.

### 4.2. Phenylalanine Metabolism

According to the results of our study, the most perturbed pathway in CTE patients was phenylalanine metabolism. Phenylalanine is an essential aromatic amino acid involved in the synthesis of tyrosine and catecholamines. In the CTE patients, level of phenylalanine increased by more than two-fold compared to the control subjects. It has been reported that a high level of phenylalanine can disrupt mood, impair sustained attention [49], and increase depressive symptoms [50], which are all common symptoms of CTE. Although not specifically characterized, deposition of amyloid β-peptide (Aβ) is associated with pathophysiology of CTE, since CTE patients exhibit unique Aβ deposition compared to normal aging people [51]. Elevated levels of phenylalanine may accelerate the production of Aβ in CTE patients because there is evidence that the aromaticity of phenylalanine triggers protein aggregation and generation of higher-order structures, especially in β-sheet motifs [52]. This elevation of phenylalanine may be due to the interference in epinephrine and norepinephrine neurotransmitter pathways following repetitive brain trauma (RBT) [53], which is commonly experienced by many CTE patients during their sports career. The elevation may also have resulted in the elevation of its down-stream metabolite tyrosine, suggested by the higher expression of *PAH* genes in CTE patients (Figure 5).

### 4.3. Tyrosine Metabolism

The second most perturbed pathway in CTE patients is tyrosine metabolism. Tyrosine metabolism is closely associated with many aspects of brain function because the first step of this metabolism is the hydroxylation of tyrosine into dihydroxyphenylalanine (DOPA) by the enzyme tyrosine hydroxylase (TH), the rate-limiting enzyme in biosynthesis of catecholamine that act as neurotransmitters in the brain [54]. In CTE patients, elevated levels of tyrosine and low expression of *TH* were observed (Figure 5), suggesting a disruption in tyrosine degradation. The low expression of *TH* in the brain of CTE patients is probably due to the repetitive head traumas because significantly reduced expression of TH and TH-positive neurons were observed in the midbrain of mice after chronic TBI [55]. Moreover, a decrease in TH activity has also been discussed in other neurodegenerative diseases, such as AD [56] and PD [57]. Although the effect of elevated tyrosine levels on CTE pathology is still unknown, there is evidence that administration of high doses of tyrosine is detrimental to cognitive function in elderly people [58], suggesting that tyrosine in excess may contribute to cognitive impairment in CTE patients.

Another metabolic change observed in the tyrosine metabolism in CTE patients is the increased amount of norepinephrine (NE). NE is a neurotransmitter that is synthesized from dopamine by dopamine-β-hydroxlase (DBH) in the locus coeruleus (LC) in the brain [59]. However, the expression levels of *DDC* and *DBH* were lower in the brain of CTE patients compared to that of control subjects (Figure 5), suggesting the existence of an unidentified alternate mechanism causing the abnormal elevation of NE in CTE.

One of the interesting functions of NE is synthesizing neuromelanins in LC [60]. In CTE, brain injuries induce iron deposition that leads to oxidative stress and tau protein phosphorylation [61]. This suggests a compensatory increase in NE in order to synthesize more neuromelanins as they can scavenge toxic substances such as metal ions [60,62]. However, neuromelanin synthesis from NE can have detrimental effects on brain function. Even at physiological pH, NE may be easily oxidized to o-quinones even at physiological pH during the synthesis, and o-quinones can produce neurotoxic compounds when they react with cysteine or glutathione [60]. Furthermore, when released into the extracellular environment, neuromelanins can cause neurodegeneration and chronic neuroinflammation [61,63].

In line with the elevated NE, its downstream metabolite vanylglycol, also known as 3-methoxy-4-hydroxyphenylglycol (MHPG), as well as its producing enzyme catechol-*O*-methyltransferase (COMT), were found in higher amounts in the CTE group. While the relationship between elevated level of NE and CTE pathology is equivocal, an elevated level of MHPG has been reported to be associated with higher accumulation of p-tau and memory deficits [64], the major neuropathological features of CTE [1,5,9,10]. In addition, elevated levels of NE and MHPG have also been consistently observed in patients with advanced AD, suggesting highly active noradrenergic function despite the decreased number of cells in LC [65]. Therefore, despite the loss of neurons in LC in CTE patients, it is anticipated that the noradrenergic system can be activated abnormally as well. Our results of abnormal activation of the noradrenergic system in CTE can be an interesting target for future study as abnormal hyperactivation of LC has been discussed as a biomarker of AD [66], and increased NE can contribute to many symptoms of CTE. For example, dysregulation in LC is associated with aggressive behavior [67], and increased NE can deprive rapid eye movement (REM) sleep which can further increase aggressiveness, lack of concentration, and memory loss [68].

Of all the metabolites detected in tyrosine metabolism, 3,4-dihydroxyphenylacetaldehyde (DOPAL) is the only metabolite that was reduced in the CTE patients. This is interesting because it has been widely reported that DOPAL is a neurotoxic metabolite of dopamine, and high level of DOPAL leads to the death of neurons in substantia nigra and accumulation of α-synuclein in PD [69,70,71]. Depigmentation and accumulation of NFTs in substantia nigra is a neuropathological feature of CTE [5,10], and many symptoms and pathological features are overlapping between CTE and PD [72]. This raises the possibility that there may be a unique mechanism that causes damage to the substantia nigra in CTE, which does not include the neurotoxic effects of DOPAL. Although the reason for the decline in DOPAL levels is currently unclear, once the mechanism of this change is elucidated in future studies, DOPAL has great potential to become a strong biomarker that can distinguish CTE from PD.

### 4.4. Limitations and Future Directions

Although we uncovered unreported biochemical pathways associated with CTE using non-targeted metabolomics, some metabolic changes could not be explained due to the lack of current knowledge in CTE. For instance, a high level of creatine, threonine, and l-glutamic acid 5-phosphate and low level of 1-pyrroline-5-carboxylate (P5C) in arginine and proline metabolism were not discussed in this study as we were unable to relate these changes to the pathophysiology or clinical features of CTE with current knowledge.

Another limitation of this study is the small sample size. Since a non-targeted metabolomics-based study has not been applied in studying CTE, it is difficult to emphasize the reliability and show consistency of this study by comparing the results of our study with other research. In order to improve the reliability of our results, expression of genes associated with significantly altered metabolic pathways was quantified to bridge the gaps in a metabolism. However, a limitation still exists as not all the metabolites in metabolism were detected via non-targeted metabolomics and some metabolic changes could not be explained even with the results of RNA sequencing. Larger sample size may therefore provide more information and increase the statistical confidence of the results.

## 5. Conclusions

We report, for the first time, the metabolic profile of postmortem brain tissue from CTE patients. Through a non-targeted metabolomics approach, we revealed detrimental consequences of astrocyte activation and significant alterations in phenylalanine and tyrosine in CTE. These findings may provide new insights into understanding the pathophysiology of CTE and offer novel therapeutic and diagnostic targets for the disease.

## Figures and Tables

**Figure 1 biomedicines-10-01718-f001:**
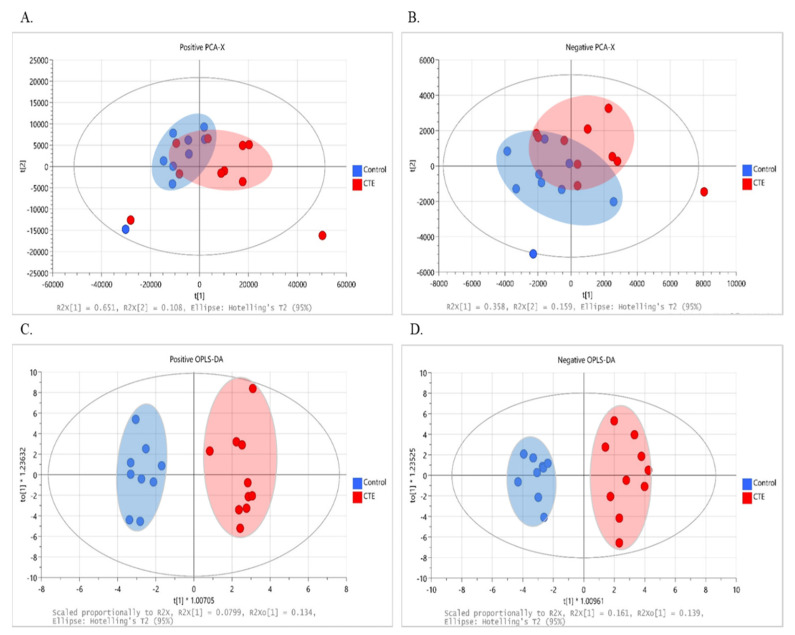
The results of multivariate analysis with 251 variables in the CTE (red) and healthy control groups (blue). (**A**) The PCA score plot at positive ionization mode. (**B**) The PCA score plot at negative ionization mode. (**C**) The OPLS–DA score plot at positive ionization mode. (**D**) The OPLS–DA score plot at negative ionization mode. PCA: principal component analysis; OPLS–DA: orthogonal projection to latent structure–discriminant analysis; CTE: chronic traumatic encephalopathy.

**Figure 2 biomedicines-10-01718-f002:**
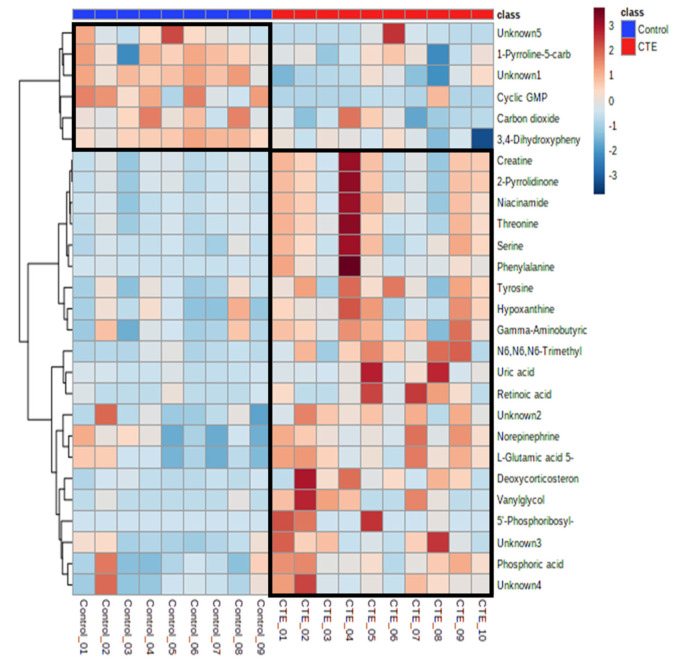
Heatmap of the 27 significantly altered metabolites in the CTE and healthy control groups. The colors blue to red indicate the concentration of metabolites from the lowest to the highest.

**Figure 3 biomedicines-10-01718-f003:**
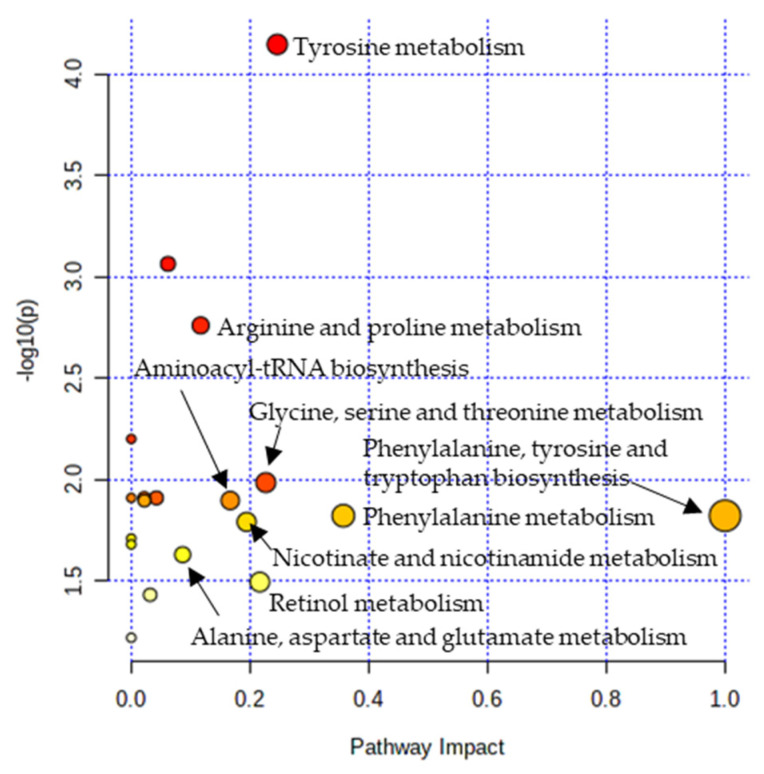
Pathway analysis showing significantly altered pathways in the CTE group. X-axis represents the pathway impact value from pathway topology analysis. Y-axis represents the −log10 (*p*) value from the enrichment analysis. The impact value is indicated by the size of the node and the significance of the pathway is indicated by the color of the node from yellow to red.

**Figure 4 biomedicines-10-01718-f004:**
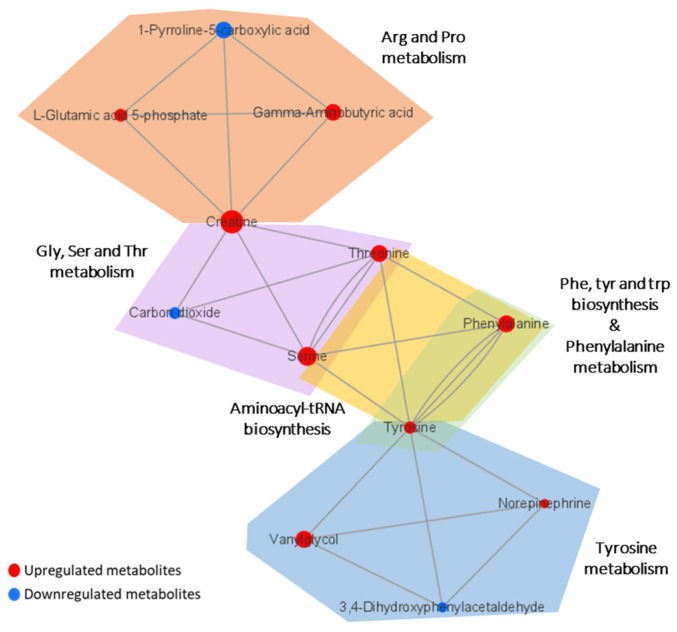
Network analysis presenting the connection between the metabolites in significantly altered pathways. The size of the nodes indicates the magnitude of the fold change. The red and blue colors indicate upregulation and downregulation, respectively.

**Figure 5 biomedicines-10-01718-f005:**
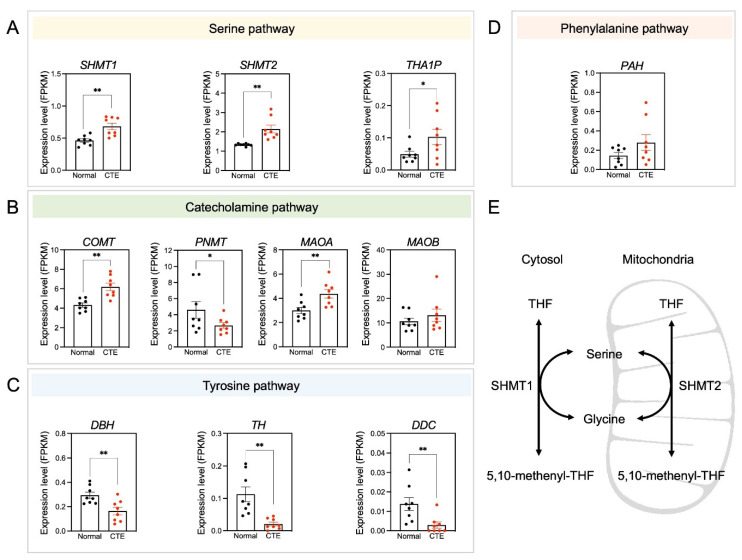
Amino acid neurotransmitter-associated genes are altered in CTE patients. (**A**) The expression levels of *SHMT1*, *SHMT2,* and *THA1P* were increased in CTE patients. (**B**) The expression levels of *COMT* and *MAOA* were increased but *PNMT* was decreased in CTE patients. (**C**) The expression levels of *DBH*, *TH,* and *DDC* were decreased in CTE patients. (**D**) The expression level of *PAH* was not different between CTE patients and normal subjects. (**E**) A scheme illustrating that serine is produced by *SHMT1* in the cytosol and *SHMT2* in the mitochondria. Significantly different from the normal subjects at * *p* < 0.05; ** *p* < 0.01.

**Figure 6 biomedicines-10-01718-f006:**
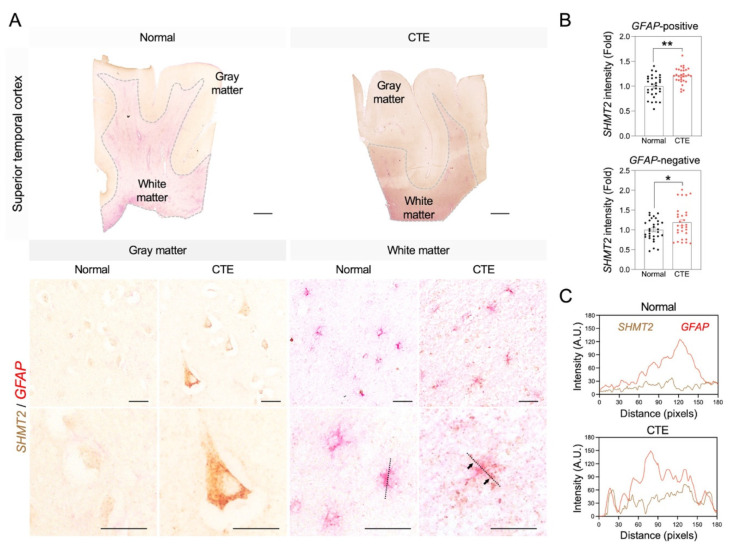
SHMT2 immunoreactivity is significantly increased in the cortex of CTE patients. (**A**) SHMT2 (brown) was co-localized with GFAP (red) in the cortical white matter of CTE patients. White dotted lines indicate the border of white matter and gray matter. Black dotted lines were drawn to analyze the colocalization of SHMT2 and GFAP. Black arrows indicate co-localization foci of SHMT2 with GFAP. Scale bars: top 2 mm, middle 20 µm, bottom 20 µm. (**B**) SHMT2 immunoreactivity was significantly increased in GFAP-positive astrocytes of the white matter and also found in GFAP-negative cells of the gray matter in CTE patients (cell counting: a total of 30 cells (10 cells/case); Normal, *N* = 3; CTE, *N* = 3). Significantly different from the normal subjects at * *p* < 0.05; ** *p* < 0.01. (**C**) Co-localization analysis showed that SHMT2 and GFAP double immunoreactivity was markedly increased in the cortical white matter of CTE patients.

**Table 1 biomedicines-10-01718-t001:** The summary of putatively identified significant metabolites.

No.	Rt	MW	Ionization Method	VIP	*p*-Value	Fold Change	Fold Change Direction	Name
1	1.06	118.93	Negative	1.71	0.0003	5.17	Down	Unknown 1
2	1.25	123.04	Positive	1.69	0.0030	0.92	Up	Niacinamide
3	9.26	204.10	Positive	1.63	0.0279	1.20	Up	Tyrosine
4	1.17	120.07	Positive	1.62	0.0172	2.44	Up	Threonine
5	1.27	132.10	Positive	1.58	0.0350	4.13	Up	Creatine
6	1.00	104.11	Positive	1.56	0.0435	2.37	Up	Gamma-Aminobutyric acid
7	1.27	86.10	Positive	1.55	0.0435	2.57	Up	2-Pyrrolidinone
8	0.95	225.99	Negative	1.45	0.0101	1.35	Up	l-Glutamic acid 5-phosphate
9	9.87	187.10	Negative	1.44	0.0220	7.01	Up	*N*6,*N*6,*N*6-Trimethyl-l-lysine
10	1.09	343.99	Negative	1.43	0.0030	0.15	Down	Cyclic GMP
11	0.94	104.03	Negative	1.41	0.0101	3.00	Up	Serine
12	12.07	329.23	Negative	1.36	0.0076	1.47	Up	Deoxycorticosterone
13	1.22	353.03	Positive	1.34	0.0452	2.89	Up	5′-Phosphoribosyl-*N*-formylglycinamide
14	1.16	219.08	Negative	1.34	0.0217	2.48	Up	Vanylglycol
15	9.80	679.51	Positive	1.33	0.0279	1.32	Up	Unknown 2
16	1.04	136.05	Positive	1.33	0.0022	2.38	Down	1-Pyrroline-5-carboxylic acid
17	13.90	335.22	Negative	1.24	0.0425	0.72	Up	Retinoic acid
18	1.35	187.00	Negative	1.23	0.0002	0.69	Down	3,4-Dihydroxyphenylacetaldehyde
19	0.94	96.92	Negative	1.23	0.0279	2311.30	Up	Phosphoric acid
20	1.08	135.03	Negative	1.21	0.0435	8.82	Up	Hypoxanthine
21	1.55	164.07	Negative	1.17	0.0006	2.47	Up	Phenylalanine
22	1.25	78.96	Negative	1.15	0.0279	1.12	Down	Carbon dioxide
23	1.00	204.00	Negative	1.14	0.0350	0.43	Up	Norepinephrine
24	0.97	134.89	Negative	1.13	0.0172	0.67	Up	Unknown 3
25	0.99	167.02	Negative	1.13	0.0030	1.27	Up	Uric acid
26	0.92	160.91	Negative	1.07	0.0350	1.79	Up	Unknown 4
27	12.57	316.25	Positive	1.06	0.0181	1.84	Down	Unknown 5

## Data Availability

All the data presented in this study are available from the corresponding author upon reasonable request.

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
