# Peer review of "Non-Targeted Metabolomics Approach Revealed Significant Changes in Metabolic Pathways in Patients with Chronic Traumatic Encephalopathy"

_biomedicines, 2022, doi:10.3390/biomedicines10071718_

Round 1

Reviewer 1 Report

General comment:

This study investigated potential metabolic pathways involved in chronic traumatic encephalopathy in a post mortem setting. The manuscript addresses a subject that is of great interest in the field of CNS disease. An extensive English language editing is required.

My suggestions for further improvements are as follows:

The abstract can be clearly presented in the following sequence: Introduction, Objective(s), Methodology, Results, Discussion and Conclusion. I find it challenging to identify these sections.

State the justification of selecting superior temporal cortex from the aspect of neuroanatomical interest. Superior temporal gyrus (STG) is the site of primary auditory cortex, auditory association cortex and multisensory integration.

 Line 65-67 – to be supported with citation(s).

"However, not only that most of the studies focused on traumatic brain injury (TBI) rather than CTE that the significance of these findings in CTE remains to be elucidated, the imaging technologies also lack specificity and sensitivity to diagnose CTE in a living person"

State the need of using a non-targeted metabolomic approach in this study.

To standardise the writing format of manufacturer for each chemical, consumables, reagent and equipment used--(brand, district, state and country)

Protocol of scoring for GFAP-positive astrocytes needs to be stated in the Methodology (with citation(s))

Figure 6A – define cortical white matter ---- cerebral cortex is the grey matter. Labelling can be added and improvised to high power photomicrographs. Photomicrographs in low power resolution does not provide any valuable information.

Author Response

REVIEWER #1

Comments and Suggestions for Authors

General comment:

This study investigated potential metabolic pathways involved in chronic traumatic encephalopathy in a post mortem setting. The manuscript addresses a subject that is of great interest in the field of CNS disease. 

1) An extensive English language editing is required.

Answer: We made English language edits throughout the manuscript.

My suggestions for further improvements are as follows:

2) The abstract can be clearly presented in the following sequence: Introduction, Objective(s), Methodology, Results, Discussion and Conclusion. I find it challenging to identify these sections.

Answer: The abstract was edited to follow the sequence as the reviewer suggested. First, we briefly described the disease and the current diagnostic challenge of CTE in the introduction. Then, we described the objective of the study, which is to advance our knowledge in CTE for better diagnosis and treatment. In the Methodology, we added the method to perform non-targeted metabolomics and explained how we determined the significant changes in the brain of study subjects. Finally, we stated the metabolic changes in CTE and their significance in understanding pathophysiology of CTE. We hope these changes improved your understanding of the study’s abstract.

3) State the justification of selecting superior temporal cortex from the aspect of neuroanatomical interest. Superior temporal gyrus (STG) is the site of primary auditory cortex, auditory association cortex and multisensory integration.

Answer: As the Reviewer suggested, we provided a rationale for analyzing the superior temporal cortex in the Materials and Methods section as identified below: “McKee et al (2015) had reported that the temporal lobe exhibits an increase of phosphorylated Tau (p-Tau) in stage IV of CTE while the region does not show prominent signals of p-Tau at the early stage of CTE. Accordingly, it is proposed that the temporal cortex reflects the pathological progression of CTE through the spreading of p-Tau after the head injury. In this context, we chose to study the temporal cortex to determine whether it shows metabolomic changes,  similar to the progressive pathological change in response to the repetitive brain trauma.”

Reference: McKee, A. C., Stein, T. D., Kiernan, P. T. & Alvarez, V. E. The neuropathology of chronic traumatic encephalopathy. Brain Pathol. 25, 350–364 (2015).

4) Line 65-67 – to be supported with citation(s).

"However, not only that most of the studies focused on traumatic brain injury (TBI) rather than CTE that the significance of these findings in CTE remains to be elucidated, the imaging technologies also lack specificity and sensitivity to diagnose CTE in a living person"

Answer: Citations were added to the sentence (Line 72-74).

5) State the need of using a non-targeted metabolomic approach in this study.

Answer: We added the justification for using non-targeted metabolomics in this study as the reviewer suggested. Non-targeted metabolomics is an unbiased approach to studying both known and unknown metabolites. Because little is known about the pathophysiology of CTE, we wanted to detect a wide range of compounds that are involved in brain functions that can help us to advance our knowledge in CTE. Because CTE is a complex disease we thought this approach could help us to better understand the dynamic interactions between different types of metabolites in the brain. This justification can be found in line 76-83 in the introduction.

6)To standardise the writing format of manufacturer for each chemical, consumables, reagent and equipment used--(brand, district, state and country)

Answer: Writing format of manufacturers in method section was standardized as reviewer suggested.

7) Protocol of scoring for GFAP-positive astrocytes needs to be stated in the Methodology (with citation(s))

Answer: We added protocols for determining SH2MT intensity in GFAP-positive and -negative cells in the 2.9 Immunohistochemistry (IHC) section.

8) Figure 6A – define cortical white matter ---- cerebral cortex is the grey matter. Labelling can be added and improvised to high power photomicrographs. Photomicrographs in low power resolution does not provide any valuable information.

Answer: We added labelling in Figure 6A to define cortical white matter and changed to high power photomicrographs.

Reviewer 2 Report

In this study, non-targeted metabolomics was performed with postmortem brain tissues of CTE patients and control subjects in order to advance our knowledge in CTE pathophysiology and identify metabolic characteristics for the early diagnosis of CTE.

Some revisions need to be done before acceptance.

1)  General Revision:

- Typography: the authors should read thoroughly their manuscript and check: 1) space between words; 2) English of some sentences

- I suggest to better discuss the CTE in the introduction

2)  Material and Methods section:

- Specify the code, commercial brand, city and country of provenance for all materials

- Please, specify the sex and the age of CTE patients and control subjects

3)  Results:

-  Figure 6 A: please add scale bar in the picture

Author Response

REVIEWER #2

Comments and Suggestions for Authors

In this study, non-targeted metabolomics was performed with postmortem brain tissues of CTE patients and control subjects in order to advance our knowledge in CTE pathophysiology and identify metabolic characteristics for the early diagnosis of CTE.

Some revisions need to be done before acceptance.

1)  General Revision:

Typography: the authors should read thoroughly their manuscript and check: 1) space between words; 2) English of some sentences

- I suggest to better discuss the CTE in the introduction

Answer: We thoroughly explained CTE in the introduction by explaining the history of the term CTE, clinical symptoms, neuropathology, diagnostic methods, as well as the current limitations of diagnosing CTE.

2)  Material and Methods section:

- Specify the code, commercial brand, city and country of provenance for all materials

Answer: Manufacturer information of reagents and equipment used in this study were specified.

- Please, specify the sex and the age of CTE patients and control subjects

Answer: The sex and the age of CTE patients and control subjects are specified in supplementary table 1.

3)  Results:

-  Figure 6 A: please add scale bar in the picture

Answer: We added scale bars in Figure 6A.